# Drug Development in Tissue-Agnostic Indications

**DOI:** 10.3390/cancers13112758

**Published:** 2021-06-02

**Authors:** Pauline du Rusquec, Christophe Le Tourneau

**Affiliations:** 1Department of Drug Development and Innovation (D3i), Institut Curie, 75005 Paris, France; pauline.durusquec@curie.fr; 2INSERM U900, 92210 Saint-Cloud, France; 3Faculty of Medicine, Paris-Saclay University, 78180 Montigny le Bretonneux, France

**Keywords:** cancer, precision medicine, molecularly targeted agents, study endpoint, progression-free survival ratio, time to progression

## Abstract

**Simple Summary:**

The primary goal of any intervention in oncology is to improve overall survival and/or quality of life. The gold-standard approach to demonstrate that this goal has been achieved is a randomized controlled clinical trial. A better understanding of cancer biology has led to the molecular segmentation of cancer, with some molecular alterations occurring across cancer types. Whereas the ancestral paradigm of drug development has evaluated new drugs in specific cancer types, the development of drugs targeting rare molecular alterations across cancer types has challenged this ancestral paradigm. Novel clinical trial designs have emerged to address this new paradigm, including basket trials and precision medicine trials that use each patient as his/her own control to assess the efficacy of a new treatment. We review here the opportunities and challenges of these new clinical trial designs.

**Abstract:**

A better understanding of cancer biology has led to the development of targeted therapies specifically designed to modulate an altered molecular pathway in the cancer cells or their microenvironment. Despite the identification of molecular targets across cancer types, most of targeted therapies were developed per cancer type. In this ancestral paradigm, randomization was the gold-standard approach for market access. Randomization of large patient populations was feasible for drugs developed in common cancer types but more challenging in rare cancer types. The traditional paradigm of drug development in oncology was further challenged by the ever-expanding molecular segmentation of cancer with ever-smaller subgroups of patients who might benefit from specific targeted therapies or immunotherapies and the identification of molecular alterations against which drugs may be effective across cancer types. In this novel drug development paradigm, novel ways of evaluating the efficacy of drugs are highly needed in these small patient populations. One approach is to use each patient as his/her own control by comparing the efficacy of a drug to the efficacy of prior treatments received. This approach allows to overcome patient heterogeneity, especially in a tissue-agnostic drug development paradigm.

## 1. Introduction

Little has changed over the last decades in terms of methods to analyze the benefits of anticancer drugs. In the literature, the first clinical trials designed to evaluate their efficacy used soft criteria such as “clinical improvement,” “reduction in the size of organ,” “sense of well-being,” and “duration of remission” [1,2,3]. These terms, however, mirror the more robust ones used today, including clinical benefit, response rate, quality of life, and duration of response. 

In contrast, our understanding of cancer biology has tremendously increased over the last decades. Research is ongoing to refine the search of biomarkers of efficacy to immunotherapy and molecularly targeted agents [4,5]. Whereas drug development used to be performed per cancer type, this new knowledge has led to the identification of molecularly defined entities for several cancer types. Drug development, that usually only considered the primary origin of the tumor, eventually included molecular targets present in the disease. Lung cancer is a striking example. Whereas EGFR mutations are quite frequent (10–20% of Caucasian patients and 50% of Asian patients), several other actionable genomic alterations are less frequent, including KRAS G12C mutation (13%), ALK translocation (5%), HER2 mutation (3%), MET activation (3%), BRAF V600 mutation (2%), and ROS1 translocation (1%) [6,7,8,9]. 

Some molecular alterations exist across cancer types, such as microsatellite instability (MSI) and NTRK fusions. Whereas MSI is common in patients with colorectal and endometrial carcinomas, MSI is infrequent in other cancer types [10]. NTRK fusions occur in 0.3% of all cancer patients, although present in most of patients with secretory breast carcinoma and mammary analogue secretory carcinoma of the salivary gland [11]. Although effective therapies exist against these molecular alterations, their very low prevalence has challenged the ancestral paradigm of drug development per cancer type. Indeed, given the very small number of patients in most of cancer types, traditional randomized clinical trials cannot be run within a reasonable period of time. 

There is therefore an urgent need for novel clinical trials designed for very small patient populations. We aim here to provide a historical perspective of endpoints that have been used in oncology to evaluate the efficacy of new drugs and discuss the potential of using each patient as his/her own control in small patient populations. 

## 2. Efficacy Endpoints: A Historical Perspective

Tumor shrinkage has initially been used to reflect treatment efficacy. Inspired by the work conducted by the Breast Cancer Task Force on breast cancer [12] and the Union for International Cancer Control’s [13], the World Health Organization (WHO) published the first classification using quantified criteria to evaluate response to treatment in 1979 [14]. The sum of the products of the two longest perpendicular dimensions of all measurable lesions was followed over time. Non-measurable lesions were only reported. Response to treatment was classified into complete response (disappearance of all target lesions) and partial response (reduction of more than 50% in the sum of the target lesions). Any target lesion with at least 25% increase in the sum of the two perpendicular dimensions met the definition of “progressive disease.” The high intra-observer variability in the application of the WHO criteria led to the definition of the Response Evaluation Criteria in Solid Tumors (RECIST) in 2000 [15]. In RECIST, a maximum of 10 target lesions had to be followed, including a maximum of 5 per organ. Target lesions had to measure at least 10 mm. The biggest change with the WHO criteria was that RECIST was unidimensional instead of bidimensional. A partial response was defined as at least a 30% decrease in the sum of the largest diameters of the target lesions, whereas progressive disease was defined as at least a 20% increase from the nadir in the sum of the largest diameters of the target lesions or the occurrence of a new target. RECIST was further amended into RECIST1.1 by further reducing the number of target lesions to a total of five and a maximum of two per organ [16]. Only lymph nodes with the smallest diameter exceeding 15 mm could be considered as target lesions. 

New criteria have been published for patients treated with immunotherapy, since these patients sometimes experience atypical responses such as pseudo-progression (response following disease progression) [17]. Several criteria were proposed during the last decade, including immune related Response Criteria (irRC) [18], immune-related RECIST (irRECIST) [19], immune RECIST (iRECIST) [20], and immune-modified RECIST (imRECIST) [19]. They all share the common concept of “unconfirmed progressive disease,” allowing the continuation of treatment post-progression in case of suspected pseudo-progression. In that latter situation, patients continue treatment until the next tumor evaluation that either confirms disease progression (and treatment failure) or suggests pseudo-progression [21]. However, RECIST remains the reference in clinical research, with immune-related criteria being used as secondary endpoints. RECIST1.1 serves as the basis to evaluate surrogate endpoints for survival that include disease progression in their definition. These endpoints and their definitions are summarized in Table 1.

## 3. Ancestral Paradigm of Drug Approval in Oncology

The demonstration of a benefit of a new treatment has been based on randomization versus standard of care to avoid selection bias. The benefit is, however, only restricted to the patient population that was selected for the trial, excluding a substantial proportion of real-life patients. The gold-standard achievements for market authorization include an improvement in overall survival and/or quality of life. To speed up drug development in oncology, the use of surrogate endpoints for overall survival has been advocated. Overall response rate and survival-based endpoints such as progression-free survival or disease-free survival have been used to obtain an early readout of treatment efficacy.

In 2018, the FDA published an exhaustive list of surrogate endpoints that can be used to foster the development of treatments in onco-hematology [22]. These surrogate endpoints are all based on RECIST1.1. From an analysis of 107 anticancer drugs across 188 indications, Chen et al. showed that the use of overall response rate and progression-free survival as surrogate endpoints for overall survival would have saved 19 and 11 months for market authorization, respectively [23]. The use of these surrogate endpoints remains debated since correlations between these endpoints and overall survival have been inconstantly demonstrated. In addition, the correlations made in these studies only apply to the drugs that were used in the included studies. As an example, progression-free survival was shown to be a valid surrogate endpoint for overall survival in studies evaluating the combination of chemotherapy with an antiangiogenic agent in metastatic colorectal cancer patients [24]. Whether this correlation would hold true when including EGFR-targeting combinations remains to be determined. Kim et al. analyzed FDA marketing authorizations between 2008 and 2012, and reported that 36 out of the 54 molecules (67%) were approved based on an improvement of a surrogate endpoint for overall survival [25]. Only five of these drugs were eventually shown to produce an overall survival benefit. 

Overall response rate and progression-free survival have clearly been shown not to correlate with immunotherapy efficacy [26]. Overall response rates with immunotherapy as single agents are consistently low and progression-free survival is consistently short because most of patients actually do not respond to immunotherapy. Now, responding patients usually experience durable responses.

Based on the assumption that the reduction of the tumor burden is associated with a treatment benefit in a given patient, several predictive markers of tumor response have been evaluated. The LDH serum level, for example, has been shown to inversely correlate with the PFS under treatment [27]. Its normalization has been shown to predict a benefit to immunotherapy in melanoma [28,29]. The benefit of adding bevacizumab as a first-line chemotherapy in Chinese patients with metastatic colorectal cancer has been shown to correlate with the level of LDH in the sera of patients at baseline [30]. Similar results were reported with the albumin concentration, the neutrophil-to-lymphocyte ratio, the number of metastatic sites, the level of circulating tumor DNA, non-coding microRNAs, etc. [31]. 

## 4. Impact of the Molecular Segmentation of Cancer 

The molecular segmentation of cancer has challenged the ancestral paradigm of drug development. One compelling example is the development of crizotinib in patients with non-small cell lung cancer (NSCLC) with ALK rearrangement into an EML4-ALK fusion protein [32]. Kwak et al. conducted a phase I expansion cohort of crizotinib in 82 patients with pretreated ALK positive NSCLC patients, for which they had to screen 1500+ patients. The overall response rate was as high as 57% [33]. Following these results, a randomized controlled phase III trial against chemotherapy had to be conducted in Europe for market authorization. A total of 4967 patients had to be screened to randomize 347 patients. The benefit of crizotinib over chemotherapy in this patient population was confirmed with an overall response rates of 65% versus 20% (*p* < 0.001), a median PFS of 7.7 months versus 3.0 months (*p* < 0.001) [34]. One might ask whether a randomized trial was necessary, given the exceptional results obtained in the phase I trial and the high number of patients to be screened. 

A breakthrough occurred in the history of oncology when, for the first time, the FDA granted approval of an anticancer drug based on a molecular alteration in a tissue-agnostic way. Ironically, this first drug was not a molecularly targeted agent but pembrolizumab, an immunotherapy mainly acting on T cells in patients with high microsatellite instability (MSI-H) or a MMR deficiency (dMMR). [35]. This approval was based on the results of five nonrandomized phase I or II clinical trials (KEYNOTE-012, -016, -164, -028, -158) conducted in 149 patients with MSI-H/dMMR tumors. These results were obtained by prospective or retrospective analyses of the aforementioned clinical trials. Endometrial, colorectal, and gastric cancers accounted for 30%, 20%, and 20% of MSI-H/dMMR tumors, respectively, whereas all other tumor types represented less than 5% of the whole cohort [10]. The high efficacy of pembrolizumab in this patient population was confirmed in a randomized trial in the subgroup of patients with metastatic colorectal cancers [36,37].

More recently, larotrectinib [38] and entrectinib [39] also obtained marketing authorization for tumors with NTRK fusion in a tissue-agnostic way. The TRK (tropomyosin receptor kinase) family of transmembrane receptors is composed of three members (TrkA, TrkB, TrkC), encoded, respectively, by three NTRK (neurotrophic receptor tyrosine kinases) genes (NTRK1, NTRK2, and NTRK3). The presence of this kinase leads to cell differentiation and is therefore a potential oncogenic driver [40]. Their deregulation is driven by a chromosomal rearrangement resulting in the fusion of an NTRK gene with one of the multiple known partners (e.g., ETV6, AFAP1, EML4). These fusions lead to permanent activation of the TRK receptors involved and to the development of cancer [41].

Larotrectinib was approved based on the review of 3 phase I or II trials (LOXO-TRK-14001, SCOUT, and NAVIGATE) conducted in 55 adults and children with cancer harboring a NTRK fusion. The ORR was 75% [42]. The efficacy of entrectinib was demonstrated in three phase I or II trials (STARTRK-1 [43], STARTRK-2 [44], and ALKA-372-001 [45]) in patients with advanced cancer harboring a fusion of NTRK1/2/3, ROS1, or ALK. The pooled analysis of these 3 trials on 54 patients with a NTRK fusion showed an objective response rate of 57%. Median duration of response and progression-free survival were 10.4 months and 11.2 months, respectively [46]. The presence of a NTRK 1/2/3 fusion is a rare oncologic event which is present in less than 1% of patients with solid tumors [47]. 

Finally, the FDA granted marketing approval in 2020 for pembrolizumab in advanced cancers with a high Tumor Mutational Burden (TMB) calculated from the Foundation using 1 assay with a threshold of 10 mutations/Mb [48]. In this large study including patients with all types of refractory cancer, 13% of patients had a high TMB. This subgroup particularly benefited from pembrolizumab with an ORR of 29% versus 6% for the non-TMB-high subgroup.

None of these drugs were approved in Europe in a tissue-agnostic way because of the lack of comparator. It has therefore become urgent to develop new methods for assessing the efficacy of drugs in small patient populations, especially in patients with very rare druggable molecular alterations, sometimes relevant across cancer types.

## 5. The PFS Ratio as a Novel Endpoint to Overcome Patient Heterogeneity

Using each patient as his/her own control is appealing, since it allows overcoming patient heterogeneity, especially when patients with various cancer types are eligible for a drug based on a molecular alteration. A way of doing this is to compare the efficacy of a drug that has been given during a period of time to the efficacy of another drug that has been given at another period of time. The comparison of the progression-free survivals on both periods of time can be done to compare the efficacy of both drugs by calculating the PFS ratio for a given patient. However, the use of this endpoint relies on three requirements. First, it is assumed that tumor growth is linear over time. This assumption probably holds true in the recurrent and/or metastatic setting when drugs are given for relatively short periods of time, knowing it takes years for a tumor to become macroscopic. Second, the timing of efficacy assessments should be the same for both drugs. Different timings of efficacy assessments might lead to over- or underestimating the proportion of patients with favorable PFS ratio. Third, the methods used to assess the efficacy should be the same for both drugs. Different methods for efficacy assessments might also lead to over- or underestimating the proportion of patients with a favorable PFS ratio. The principle of the PFS ratio is explained in Figure 1. A PFS ratio exceeding 1 indicates a higher efficacy of the second drug over the first one, whereas a PFS ration below 1 suggests the contrary. This endpoint is also meaningful for patients, since the longer they are on a drug, the better it means for them. It is also expected that cancer in the recurrent and/or metastatic setting has a decreasing efficacy of treatments over time.

Von Hoff was the first to propose the PFS ratio in 2010 as an efficacy endpoint in precision medicine in a tissue-agnostic clinical trial [49]. In this study, a PFS ratio exceeding 1.3 indicated a benefit of the new treatment that matched an identified molecular alteration over the last treatment received. In their study, 27% of patients who received matched therapy had a PFS ratio exceeding 1.3, suggesting that the strategy benefited 27% of treated patients.

## 6. Review of Clinical Trials Using Each Patient as His/Her Own Control to Assess Drug Efficacy

Given the existence of actionable molecular alterations across cancer types, precision medicine trials beside basket trials have been set up to evaluate the efficiency of matching drugs with molecular alterations in a tissue-agnostic way [50]. Whereas only a few clinical trials were randomized, such as SHIVA01 [51] and MPACT [52], most of the other precision medicine trials were not randomized. These latter trials used patients as their own controls to evaluate the efficiency of the approach (Table 2) [49,51,53,54,55]. Cross-over at disease progression was allowed in SHIVA01, which was unable to also use each patient as his/her own control in the subgroup of patients who did cross over from one treatment arm to another [56]. The study endpoint was the PFS ratio in all these trials (Table 2). The most used threshold to define the efficacy of the new treatment was a PFS ratio exceeding 1.3. 

The proportion of patients evaluable for the PFS ratio ranged from 13% to 36%, except in the von Hoff’s study, in which 62% of patients were evaluable. In this latter study, patients could receive not only molecularly targeted therapy but also chemotherapy, hormone therapy, and even drugs given outside oncology such as metformin [46]. The global low proportion of patients who could receive matched therapies and be evaluable for the PFS ratio include those with a lack of druggable molecular alterations identified and difficulty assessing PFS1 on last prior treatment.

The proportion of patients with a PFS ratio exceeding 1.3 in these studies varied from 25% to 61%, suggesting the efficiency of the strategy in a limited subgroup of patients. None of the studies reported the proportion of patients with, a contrario, a PFS ratio below 0.7. The SHIVA01 might, however, shed some light on these results. Indeed, 37% and 61% of patients who could cross over had a PFS ratio exceeding 1.3 [53]. However, the overall result of the comparison between the two arms of SHIVA01 did not show any statistical difference in terms of progression-free survival. The interpretation of this is that only the treatment algorithm used in SHIVA01 benefited a subgroup of patients but might have been detrimental in another one. It is therefore critical that clinical trials using the PFS ratio as an endpoint report all ratios and not only PFS ratios exceeding 1.3.

It is striking to see that the median PFS on matched therapy was short in these studies, ranging from 2.0 to 3.7 months. More importantly, the median PFS on last prior therapy was only reported in SHIVA01. Since most of patients in these trials had a PFS ratio below 1.3, it is anticipated that median PFS on last prior treatment was even shorter than the median PFS on matched therapy. 

The results obtained in these precision medicine trials highly depend on the treatment algorithms used to allocate treatments to patients. Whereas some molecular alterations might be relevant across cancer types, others are not. As an example, vemurafenib does not produce high efficacy in BRAF V600 mutated colorectal cancer with an overall response rate of 5% [57], whereas vemurafenib is highly effective in BRAF V600 mutated non-small cell lung cancer [58] and melanoma [59] with an overall response rate around 45%. These differences in efficacy most likely rely on the fact that current treatment algorithms ignore co-existing molecular alterations that might be involved in resistance mechanisms, as well as tumor heterogeneity. To illustrate the example of vemurafenib, preclinical research has shown that, in BRAF-mutated colon cancer, there is a positive feedback on the EGFR pathway [60]. This might explain the primary resistance of these tumors to vemurafenib, suggesting the rationale for a dual anti-BRAF plus anti-EGFR blockade. Indeed, this combo provides a slightly higher benefit with an ORR of 15–19% (combination vemurafenib + cetuximab [61] and encorafenib + cetuximab [62], respectively).

## 7. Conclusions

A better understanding of cancer biology has led to the emergence of very small patient populations characterized by very rare molecular alterations. A second layer of difficulty is the identification of molecular alterations relevant across cancer types such as NTRK fusions and MSI. The combination of rare molecular alterations across cancer types has challenged the ancestral drug development paradigm. Whereas we clearly consider that randomized controlled clinical trials should remain the standard of care whenever feasible, even in the context of tissue-agnostic drug development, we acknowledge the urgent need of developing new methods for small patient populations. The use of the PFS ratio is clearly one of the solutions, although its use needs to be refined, especially by ensuring that both timings and methods for efficacy assessments are the same during all treatment periods. This is exactly how the SHIVA02 ongoing precision medicine trial (NCT01771458) has been designed. Improvements in the use of this new endpoint are clearly needed. Clues include the use of continuous values of the sum of target lesions instead of using the categorical RECIST criteria.

## Figures and Tables

**Figure 1 cancers-13-02758-f001:**
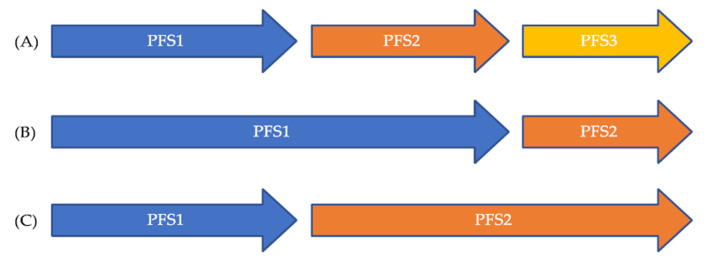
Illustration of the PES ration. (**A**) Expected natural history of cancer in the recurrent and/or metastatic setting with decreasing efficacy of treatments over time; (**B**) PES ratio < 1 suggesting a decreased efficacy of the second drug over the first; (**C**) PES ratio > 1 suggesting a benefit from the new treatment over the first.

**Table 1 cancers-13-02758-t001:** Survival-based endpoints.

Endpoint	Start	Event	Censored Events	Pros	Cons
PFS (Progression-Free Survival)	Randomization	Disease progression; Death	Last date of radiological assessment	Earlier read out Effect on survival not diluted by subsequent therapies	Death might not be related to disease but to comorbidities
TTP (Time to Progression)	Randomization	Disease progression	Death	Earlier read out Effect on survival not diluted by subsequent therapies	Does not take into account death related to treatments
TTF (Time to Treatment Failure)	Randomization	Disease progression; Adverse events; Patient choice; Death	-	Earlier read out Effect on survival not diluted by subsequent therapies Best indicator of the treatment’s specific tolerance	Does not completely reflect the duration of treatment efficacy, since patients might still be responding to treatment although they had an adverse event or decided to stop treatment Death might not be related to disease but to comorbidities
FFS (Failure-Free Survival)	The first day of treatment	Disease progression	Adverse event Death	Earlier read out Effect on survival not diluted by subsequent therapies Best indicator of the treatment’s specific efficacy	Death and adverse events related to treatments are not taken into account
DOR (Duration of Response)	Date of response	Disease progression	Death	Measures duration of response that might be of importance for some treatments such as immunohterapies	Only applies to the responding patient population
OS (Overall Survival)	Randomization	Death	Last date patient seen alive	Direct measure of clinical benefit, Easily measured, Gold standard endpoint	affected by post-progression and cross over therapies; require prolonged Follow up; includes non-cancer related deaths

**Table 2 cancers-13-02758-t002:** Selected clinical trials using each patient as his/her own control to assess efficacy.

Study	Von Hoff’s Study [49]	SHIVA01 [51,56]	MOSCATO-01 [53]	I-PREDICT [54]	WINTHER [55]
Threshold used for PFS ratio	1.3	1.3	1.3	1.3	1.5
No. of patients included	106	741	1035	149	303
No. of patients treated with matched therapy (%)	66 (62%)	170 (23%)	199 (19%)	73 (49%)	107 (35%)
No. of evaluable patients for the PFS ratio (%)	66 (62%)	95 (13%)	193 (19%)	53 (36%)	107 (35%)
Proportion of patient with a PFS ratio >1.3	27%	37% ^1^ 61% ^2^	33%	45%	25%
Median PFS1 (months)	-	2.0 ^1^ 2.3 ^2^	-	-	-
Median PFS2 (months)	-	2.1 ^1^ 2.8 ^2^	2.3	3.7	2.0

^1^ Patients crossing from physician’s choice to matched therapy; ^2^ Patients crossing over from matched therapy to physician’s choice.

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
