# Peer review of "Drug Development in Tissue-Agnostic Indications"

_cancers, 2021, doi:10.3390/cancers13112758_

Round 1
Reviewer 1 Report
This is an excellent and interesting study. I have follwoing specific comments;
- authors state 'Overall response rates with immunotherapy as single agents are consistently low and progression-free survival short, because most of patients actually do not respond to immunotherapy. Now, responding patients usually experience durable responses'. In such sentences, authors did not explain the rational very clearly such as why most of the patients do not respond to immunotherapy?
- LIne 301: '..... molecular alterations that might be involved in resistance mechanisms, as well as tumor heterogeneity'. what are these alterations and resistance mechanisms. Actually authors hardly discuss resistance mechanisms.
Author Response
Tuesday, May 26, 2021
Dear Reviewers,
Please find attached the revised manuscript entitled: “Drug development in tissue-agnostic indications" be considered for publication as a review for the Special Issue: “The Development of Anti-cancer Agents” in Cancers, which was invited by Prof. Helmout Modjtahedi, Guest Editor of Cancers.
As requested, we have responded to the comments of the 4 reviewers, whom we thank for their involvement.
We confirm that this work is original and has not been published elsewhere, nor is it currently under consideration for publication elsewhere.
We have no conflicts of interest to disclose. Please address all correspondence concerning this manuscript at christophe.letourneau@curie.fr
Thank you very much for your consideration and help in the process.
Sincerely yours,
Pauline du Rusquec, MD
Christophe Le Tourneau, MD, PhD
Reviewer 1
Point 1
- authors state 'Overall response rates with immunotherapy as single agents are consistently low and progression-free survival short, because most of patients actually do not respond to immunotherapy. Now, responding patients usually experience durable responses'. In such sentences, authors did not explain the rational very clearly such as why most of the patients do not respond to immunotherapy?
Response 1
We thank the Reviewer for this comment. Biomarker of response to immunotherapy is a huge area of research, that, we believe, is out of the scope of this review.
Point 2
- LIne 301: '..... molecular alterations that might be involved in resistance mechanisms, as well as tumor heterogeneity'. what are these alterations and resistance mechanisms. Actually authors hardly discuss resistance mechanisms.
Response 2
We thank the Reviewer for this comment. In order to illustrate potential resistance mechanisms that are not the scope of this review, we added the following section with the example of resistance to BRAF inhibitors in patients with tumors harboring BRAF V600E mutations:
To illustrate the example of vemurafenib, pre-clinical research has shown that in BRAF-mutated colon cancer, there is a positive feedback on the EGFR pathway [56]. This might explain the primary resistance of these tumors to vemurafenib, suggesting the rationale for a dual anti-BRAF plus anti-EGFR blockade. Indeed, this combo provides a slightly higher benefit with an ORR of 15-19% (combination vemurafenib + cetuximab [57] and encorafenib + cetuximab [58] respectively).
Reviewer 2 Report
In this manuscript by du Rusquec and Le Tourneau, they have a discussion on the usage of each patient as his/her own control by comparing the efficacy of a drug to the efficacy of prior treatments received. Overall, this topic should be interested to readers in this field. However, it’s not well written in the current form. I have the following comments.
- This review is relatively short. Is it a mini-review? If not, the authors need to expand the content of this review.
- This review only includes 56 references. This size is relatively small in regards to a review. They should include more references.
- There are a lot of grammar mistakes. For example, in Line 84, they stated “Efficacy endpoints: an historical perspective”. It should be “a”, rather than “an”. The manuscript needs language editing service to polish the language and correct all the grammar mistakes.
- The authors stated that “Ironically, this first drug was not a molecularly targeted agent but pembrolizumab that is an immune checkpoint inhibitor targeting PD1 in patients with high microsatellite instability (MSI-H) or an MMR deficiency (dMMR)”. This statement is confusing. Although Pembrolizumab is used for immunotherapy, it can target PD1 pathway. Meanwhile, this statement reads awkward. Please rephrase it.
- Table 1 lists several survival-based endpoints. They should also add the pros and cons of each endpoint.
- The whole manuscript is not logically written. They should reorganize it.
Author Response
Tuesday, May 26, 2021
Dear Reviewers,
Please find attached the revised manuscript entitled: “Drug development in tissue-agnostic indications" be considered for publication as a review for the Special Issue: “The Development of Anti-cancer Agents” in Cancers, which was invited by Prof. Helmout Modjtahedi, Guest Editor of Cancers.
As requested, we have responded to the comments of the 4 reviewers, whom we thank for their involvement.
We confirm that this work is original and has not been published elsewhere, nor is it currently under consideration for publication elsewhere.
We have no conflicts of interest to disclose. Please address all correspondence concerning this manuscript at christophe.letourneau@curie.fr
Thank you very much for your consideration and help in the process.
Sincerely yours,
Pauline du Rusquec, MD
Christophe Le Tourneau, MD, PhD
Reviewer 2
Point 1
This review is relatively short. Is it a mini-review? If not, the authors need to expand the content of this review.
Response 1
We understand the Reviewer’s comment. The field of tissue-agnostic drug development is quite new and there is not that much literature on methods to address this question. This is why our review might be shorter than expected. Now, we believe that we nicely covered the question including summary tables.
Point 2
This review only includes 56 references. This size is relatively small in regards to a review. They should include more references.
Response 2
Please see our response just above.
Point 3
There are a lot of grammar mistakes. For example, in Line 84, they stated “Efficacy endpoints: an historical perspective”. It should be “a”, rather than “an”. The manuscript needs language editing service to polish the language and correct all the grammar mistakes.
Response 3
We are really sorry for this comment and corrected all grammar mistakes.
Point 4
The authors stated that “Ironically, this first drug was not a molecularly targeted agent but pembrolizumab that is an immune checkpoint inhibitor targeting PD1 in patients with high microsatellite instability (MSI-H) or an MMR deficiency (dMMR)”. This statement is confusing. Although Pembrolizumab is used for immunotherapy, it can target PD1 pathway. Meanwhile, this statement reads awkward. Please rephrase it.
Response 4
We thank the Reviewer for this comment and modified the sentence as follows:
“Ironically, this first drug was not a molecularly targeted agent but pembrolizumab but an immunotherapy mainly acting on T cells in patients with high microsatellite instability (MSI-H) or an MMR deficiency (dMMR)”.
Point 5
Table 1 lists several survival-based endpoints. They should also add the pros and cons of each endpoint.
Response 5
We thank the Reviewer for this comment and added pros and cons of each endpoint into Table 1.
Point 6
The whole manuscript is not logically written. They should reorganize it.
Response 6
We are not sure how to interpret this since we think the flow is quite logical indeed.
Reviewer 3 Report
We read with interest the current review.
Targeted agents and immunotherapy have revolutionized the treatment landscape of several hematological and solid tumors by producing unprecedented algorithm shifts in a relatively short period of time. However, immune checkpoint inhibitors (ICIs) have been suggested to be effective in approximately one third of all cancer patients, with the antitumor activity of immunotherapy varying among different malignancies. Thus, the identification of potential responders has recently become one of the key challenges in medical oncology, since there is an urgent need to develop reliable biomarkers that could guide clinicians in patient selection. In fact, several predictors of response to ICIs have been tested and evaluated, three of whom have been approved by the United States Food and Drug Administration (FDA): programmed death ligand 1 (PD-L1), tumor mutational burden (TMB), and microsatellite instability / defective mismatch repair (MSI/dMMR). Notably enough, all these predictors present notable differences in terms of methodology and specificity as well as strengths and weaknesses.
Two years after the landmark approval of PD-L1 as predictive biomarker in non-small cell lung cancer (NSCLC), pembrolizumab was approved by the FDA for the treatment of patients with MSI-high (MSI-H)/dMMR advanced solid tumors in 2017.
The manuscript is quite well written and organized. English could be improved.
Tables are comprehensive and clear.
The introduction explains in a clear and coherent manner the background of this study.
We suggest the following modifications:
- Although the authors correctly included important papers in this setting, we believe a couple of studies should be cited within the introduction (PMID: 33535621 ; PMID: 31738428 ), only for a matter of consistency. We think it might be useful to introduce the topic of this interesting study.
- The authors correctly addressed the role of ntrk. However, the authors should expand this section.
Notably enough, NTRK1, NTRK2, and NTRK3 gene fusions have been suggested to act as oncogenic drivers in a range of solid tumors, including gastrointestinal cancers; these fusions have been highlighted in around 1% of all pediatric and adult malignancies, with recent studies suggesting their role as promising therapeutic targets for anticancer treatment. In particular, the frequency of these fusions seems to vary from less than 1% in cancer types such as colorectal, lung, pancreatic, breast cancers, melanoma and other hematological and solid tumors, up to 25% in tumors including thyroid and gastrointestinal stromal tumors, to more than 90% in rare cancer types such as secretory breast carcinoma, mammary analogue secretory carcinoma, and congenital mesoblastic nephroma.
We believe that major revisions are needed.
The main strengths of this paper are that it addresses an interesting and very timely question and provides clear answers, with some limitations. We suggest a linguistic revision and the addition of some references for a matter of consistency. Moreover, the authors should better clarify some points and should add some details and studies, as suggested.
Author Response
Tuesday, May 26, 2021
Dear Reviewers,
Please find attached the revised manuscript entitled: “Drug development in tissue-agnostic indications" be considered for publication as a review for the Special Issue: “The Development of Anti-cancer Agents” in Cancers, which was invited by Prof. Helmout Modjtahedi, Guest Editor of Cancers.
As requested, we have responded to the comments of the 4 reviewers, whom we thank for their involvement.
We confirm that this work is original and has not been published elsewhere, nor is it currently under consideration for publication elsewhere.
We have no conflicts of interest to disclose. Please address all correspondence concerning this manuscript at christophe.letourneau@curie.fr
Thank you very much for your consideration and help in the process.
Sincerely yours,
Pauline du Rusquec, MD
Christophe Le Tourneau, MD, PhD
Reviewer 3
Point 1
Although the authors correctly included important papers in this setting, we believe a couple of studies should be cited within the introduction (PMID: 33535621 ; PMID: 31738428 ), only for a matter of consistency. We think it might be useful to introduce the topic of this interesting study.
Response 1
We thank the Reviewer for this comment. We added the following sentence in the introduction: “Research is ongoing to try refining the search of biomarkers of efficacy to immunotherapy and molecularly targeted agents [4,5].”
Point 2
The authors correctly addressed the role of ntrk. However, the authors should expand this section.
Notably enough, NTRK1, NTRK2, and NTRK3 gene fusions have been suggested to act as oncogenic drivers in a range of solid tumors, including gastrointestinal cancers; these fusions have been highlighted in around 1% of all pediatric and adult malignancies, with recent studies suggesting their role as promising therapeutic targets for anticancer treatment. In particular, the frequency of these fusions seems to vary from less than 1% in cancer types such as colorectal, lung, pancreatic, breast cancers, melanoma and other hematological and solid tumors, up to 25% in tumors including thyroid and gastrointestinal stromal tumors, to more than 90% in rare cancer types such as secretory breast carcinoma, mammary analogue secretory carcinoma, and congenital mesoblastic nephroma.
Response 2
We thank the Reviewer for this comment. We expanded the section on NTRK by adding the following sentences:
“The TRK (tropomyosin receptor kinase) family of transmembrane receptors is composed of 3 members (TrkA, TrkB, TrkC), encoded respectively by 3 NTRK (neurotrophic receptor tyrosine kinases) genes (NTRK1, NTRK2, and NTRK3). The presence of this kinase leads to cell differentiation and is therefore a potential oncogenic driver [40]. Their deregulation is driven by a chromosomal rearrangement resulting in the fusion of an NTRK gene with one of the multiple known partners (e.g. ETV6, AFAP1, EML4). These fusions lead to permanent activation of the TRK receptors involved and to the development of cancer [41].”
Reviewer 4 Report
It is truth that the primary goal of any intervention in oncology is to improve overall survival and/or quality of life. One approach is to use each patient as his/her own control by comparing the efficacy a drug to the efficacy of prior treatments received. The review is clinically important.
Author Response
Tuesday, May 26, 2021
Dear Reviewers,
Please find attached the revised manuscript entitled: “Drug development in tissue-agnostic indications" be considered for publication as a review for the Special Issue: “The Development of Anti-cancer Agents” in Cancers, which was invited by Prof. Helmout Modjtahedi, Guest Editor of Cancers.
As requested, we have responded to the comments of the 4 reviewers, whom we thank for their involvement.
We confirm that this work is original and has not been published elsewhere, nor is it currently under consideration for publication elsewhere.
We have no conflicts of interest to disclose. Please address all correspondence concerning this manuscript at christophe.letourneau@curie.fr
Thank you very much for your consideration and help in the process.
Sincerely yours,
Pauline du Rusquec, MD
Christophe Le Tourneau, MD, PhD
Reviewer 4
Point 1
It is truth that the primary goal of any intervention in oncology is to improve overall survival and/or quality of life. One approach is to use each patient as his/her own control by comparing the efficacy a drug to the efficacy of prior treatments received. The review is clinically important.
Response 1
Thank you for taking the time to read and analyze our paper.
Round 2
Reviewer 1 Report
Authors have addressed all the comments. I am pleased to recommend this paper for publication in Cancers.
Reviewer 3 Report
The authors extensively modified the paper according to our suggestions.
We recommend Acceptance in its current form.